# Structural regulation of halide superionic conductors for all-solid-state lithium batteries

Xiaona Li [1,2], Jung Tae Kim[2], Jing Luo[2], Changtai Zhao[3], Yang Xu[3,4], Tao Mei[4], Ruying Li[2], Jianwen Liang [2,3] ✉ & Xueliang Sun [2] ✉

Metal halide solid-state electrolytes have gained widespread attention due to their high ionic conductivities, wide electrochemical stability windows, and good compatibility with oxide cathode materials. The exploration of highly ionic conductive halide electrolytes is actively ongoing. Thus, understanding the relationship between composition and crystal structure can be a critical guide for designing better halide electrolytes, which still remains obscure for reliable prediction. Here we show that the cationic polarization factor, which describes the geometric and ionic conditions, is effective in predicting the stacking structure of halide electrolytes formation. By supplementing this principle with rational design and preparation of more than 10 lithium halide electrolytes with high conductivity over $10^{-3}$ S cm$^{-1}$ at 25 °C, we establish that there should be a variety of promising halide electrolytes that have yet to be discovered and developed. This methodology may enable the systematic screening of various potential halide electrolytes and demonstrate an approach to the design of halide electrolytes with superionic conductivity beyond the structure and stability predictions.

Lithium metal halide solid-state electrolytes (SSEs), with the formula of Li$_a$MX$_b$, have represented one of the dominant families of superionic conductors (>10$^{-4}$ S cm$^{-1}$ at 25 °C) for all-solid-state lithium batteries[1–4]. The M stands for one or multiple metal elements that can dominate the structure and facilitate the migration of lithium ions. Except for the latest developed SmCl$_3$-based chlorides[3] (large-sized Sm$^{3+}$ or La$^{3+}$) and glassified LiTaCl$_6$[2] SSEs, most Li$_a$MX$_b$ halides are developed from LiX structure. Thus, these Li$_a$MX$_b$ halides can be regarded as the distribution of cations (including lithium-ion and multivalent M cations) and vacancies in the anionic framework due to the much larger ionic radius of halogen anions[5]. The Li$_a$MX$_b$ halides can initially be divided into two types based on the different anion frameworks, including the hexagonal-close-packed (hcp) and cubic-close-packed (ccp) type anionic sublattices (Fig. 1a, b). Both hcp and ccp are close-packed with a 74% atomic packing factor[6,7]. However, the ccp structure possesses higher structural symmetry than the hcp structure. The hcp structure is described by the D$_{6h}$ point group (24 symmetry elements), while the ccp structure is the O$_h$ point group (48 symmetry elements).

When further considering the occupation of cations, the ccp stacked Li$_a$MX$_b$ halides normally possess a monoclinic structure, which is labeled as ccp-M (space group C2/m). While there are two structure types for the hcp stacked halides, namely the trigonal (hcp-T, space group P$\bar{3}$m1) and orthorhombic (hcp-O, space group Pnma) structures[8]. Here, the different occupancies of cations directly lead to the discrepancy in space-group symmetries between hcp-T and hcp-O typed Li$_a$MX$_b$ halides. All three structure types are based on octahedral coordination. Halide lithium superionic conductors with these

[1]Eastern Institute for Advanced Study, Eastern Institute of Technology, Ningbo, Zhejiang 315200, P. R. China. [2]Department of Mechanical and Materials Engineering, University of Western Ontario, 1151 Richmond St, London, Ontario N6A 3K7, Canada. [3]Solid State Batteries Research Center, GRINM (Guangdong) Institute for Advanced Materials and Technology, Foshan, Guangdong 528051, P. R. China. [4]School of Materials Science and Engineering, Hubei University, Wuhan 430062, P. R. China. ✉e-mail: liangjianwen@grinm.com; xsun9@uwo.ca

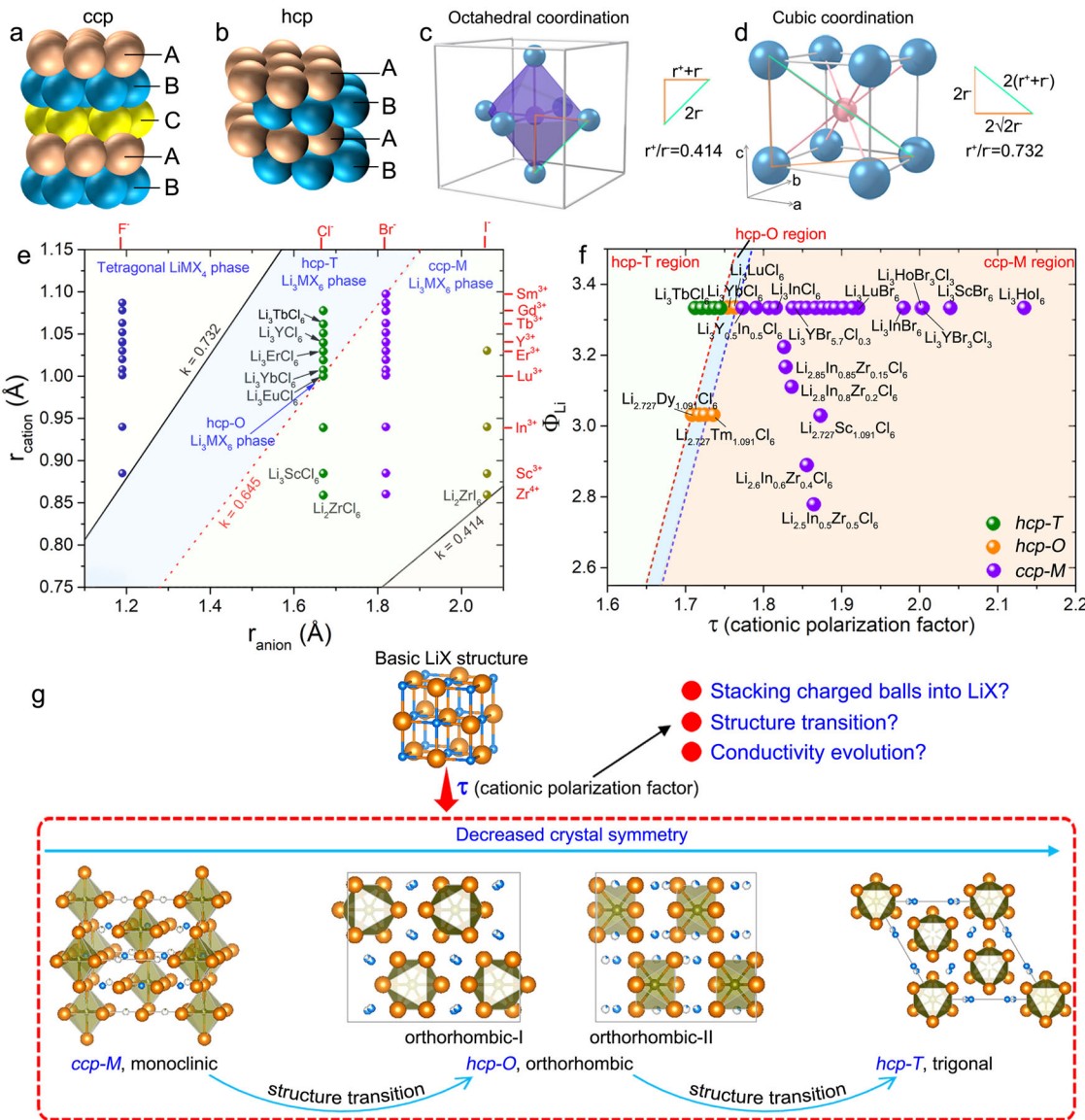

**Fig. 1 | Cationic polarization factor and its application in designing metal halide SSEs.** Schematic illustration of crystal representatives (**a**) ccp type (cubic-closed-packed anion sublattice) and (**b**) hcp type (hexagonal-closed-packed anion sublattice) metal halide SSEs. The illustration of (**c**) octahedral and (**d**) cubic coordination and the critical radius ratio. **e** Ionic radius rule and (**f**) cationic polarization factor of representative hcp-T, hcp-O, and ccp-M type $Li_aMX_b$ halides[11,12,15–18,21–27], considering the radii, Li content, metal cations, and halogen anions (see Supplementary text and Supplementary Fig. 1, Supplementary Tables 1–4 for details). **g** Schematic illustration of the evolution from basic LiCl structure to other structures and phase transition within these structures dominated by proposed τ (cationic polarization factor). ccp-M indicates the C2/m space group, hcp-T indicates the P$\bar{3}$m1 space group, and hcp-O indicates the Pnma space group.

stackings show distinctly different $Li^+$ ion migration properties. The ccp-M typed and hcp-O typed $Li_aMX_b$ halides usually possess higher $Li^+$ conductivity over $10^{-3}$ S cm$^{-1}$ at 25 °C[1,9–12]. Normally, the structural transition among those different structures as well as the induced variation of ionic conductivities can occur through substitutions or adjusting the synthesis conditions[12–14].

In the search for SSEs with high ionic conductivity, wide electrochemical stability window, and good compatibility with oxide cathodes, various hcp-T, hcp-O, and ccp-M typed $Li_aMX_b$ halides have been synthesized and investigated. However, effective guidelines for designing and preparing optimal halide SSEs are lacking, which continues as a great challenge to accelerating superionic materials discovery. The basic descriptor is the radius ratio rule that relates to the structure and composition of solid-state compounds. For $Li_aMX_b$ type halides based on octahedral sublattices, according to the law of ionic packing, they can be differentiated from other halides

based on the radius ratio of cation to anion ($k = r^+/r^-$). Two critical values of 0.414 (octahedral factor) and 0.732 (cubic factor) are obtained from the geometric information as presented in Fig. 1c, d. The octahedral sublattice structure can be theoretically stable when $0.414 < k < 0.732$. Thus, the two critical lines calculated from the radius ratio of multivalent cations and halide ions distinguish the $Li_aMX_b$ type halides from others (Fig. 1e and Supplementary Table 1). We further propose another key k value of 0.647 (Supplementary Fig. 1 and Supplementary Table 2 for detailed calculation) that separates the cubic-close-packed (ccp-M) and hexagonal-close-packed (hcp-T and hcp-O) structures in these $Li_aMX_b$ halides. This hints that the ionic radius of the cations/anions plays an important role in the stackings of the halide SSEs. Nevertheless, this method only accounts for the difference in cation/anion radii, which makes it impossible to predict the structure of halides with the same metal cations and anions (e.g. hcp-T typed $Li_3HoCl_6$ and hcp-O

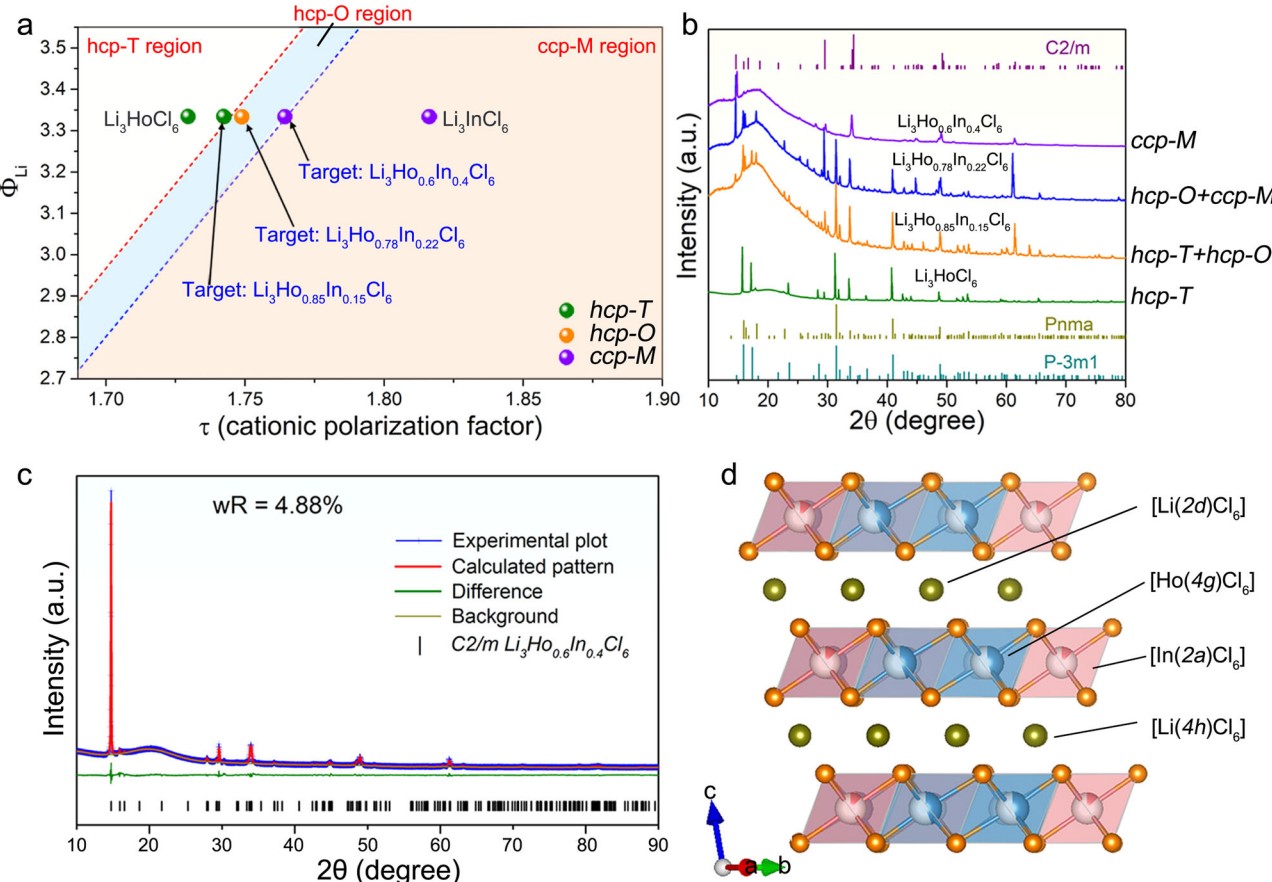

**Fig. 2 | Cationic polarization factor and structural design guide for halides through the M³⁺ regulation. a** Analysis of the cationic polarization factor for Li₃Ho₁₋ₓInₓCl₆ halides (see Supplementary Table 5 for details). **b** XRD patterns of the targeted Li₃Ho₁₋ₓInₓCl₆ samples and the standard references. **c** Rietveld refinement of XRD pattern of ccp-M typed Li₃Ho₀.₆In₀.₄Cl₆ (see Supplementary Tables 6, 7 for details). **d** Schematic illustration of the corresponding structure of ccp-M typed Li₃Ho₀.₆In₀.₄Cl₆ sample. ccp-M indicates the C2/m space group, hcp-T indicates the P3̄m1 space group, and hcp-O indicates the Pnma space group.

typed Li$_{2.727}$Ho$_{1.091}$Cl$_6$)[11], let alone give further guidance of structure regulation.

## Results

Besides the basic crystal size requirements, another significant parameter that dominates the stability of solid-state compounds is the ionic potential of ions. The ionic potential (I) is defined as the ratio of cationic charge ($z^+$) and cationic radius ($r^+$), which reflects the polarizing power of cations. The ionic potential gives an indication of the strength of the electrostatic attraction of ions with opposite charges. Aiming at a guide for the crystal structure control of the halide SSEs, we propose the "cationic polarization factor" ($\tau$) to predict the structure of Li$_a$MX$_b$ halides based on the chemical composition, which has the form

$$\tau = \frac{\Sigma\Phi cations}{\Sigma\Phi X} = \frac{\Phi Li + \Sigma\Phi M}{\Sigma\phi X} \quad (1)$$

where $\Phi_{Li}$ represents the molar content proportionally ionic potential of lithium ions, which is defined as $\Phi_{Li} = n_{Li} \times I_{Li}$; Similarly, $\Sigma\Phi_M$ and $\Sigma\Phi_X$ represent the summation of the molar content proportionally ionic potential of different ions.

Then, we can apply the proposed cationic polarization factor to distinguish and design Li$_a$MX$_b$ halide superionic conductors. The prediction of Li$_a$MX$_b$ halides stability using the cationic polarization factor requires only the chemical composition. In addition to predicting if a Li$_a$MX$_b$ material is stable as ccp-M, hcp-T, or hcp-O

structure, the cationic polarization factor also provides an estimate of the phase transition among those different structures, which further induces different Li-conducting behavior (Fig. 1f, g). Thus, the forecast based on the cationic polarization factor can constitute a useful predictive tool for those still unexplored halides and for discovering new halides with superionic conductivity.

All the reported information to our best knowledge about the Li$_a$MX$_b$ halides with octahedral sublattice was used to analyze the influence of $\tau$. The radii used here were obtained based on Shannon crystal ionic radii which are closer to the physical size of ions in a solid[15]. The molar content proportionally ionic potential of lithium ions ($\Phi_{Li}$) were plotted as a function of the $\tau$ values to show the phase map of the reported Li$_a$MX$_b$ halides as shown in Fig. 1f. There are two lines that separate the hcp-T, hcp-O, and ccp-M type halide SSEs, indicating that the $\tau$ is a promising descriptor of the crystallography structure of the Li$_a$MX$_b$ halides. A lower $\tau$ implies stronger electronic cloud extension of the cations, thus leading to a stronger impact on the halogen anion framework when the cations stack into the anion framework. Finally, this will result in a degradation of the anion framework symmetry from ccp to hcp arrangement. On the other hand, a larger $\tau$, achieved in the case of smaller M cations or larger anion framework, has less impact on the halogen anion framework, favoring the ccp structure (Fig. 1g).

Using the $\tau$ as a guide, we design specific stacking structures by controlling the compositions of the Li₃MX₆ halides. A notable starting point is Li₃HoCl₆, which possesses an hcp-T typed structure[11,13]. Increasing the $\tau$ suggests a possible route to adjust the structure to

the ccp-M typed structure via partial substituting of $Ho^{3+}$ by $In^{3+}$ (Fig. 2a), where $In^{3+}$ has a higher ionic potential. As presented, three target samples (including $Li_3Ho_{0.85}In_{0.15}Cl_6$, $Li_3Ho_{0.78}In_{0.22}Cl_6$, and $Li_3Ho_{0.6}In_{0.4}Cl_6$) were designed and synthesized to validate the trigger point of the structural transition. The $Li_3Ho_{0.85}In_{0.15}Cl_6$ sample with a small amount of $In^{3+}$ substitution induces the transition from pure hcp-T structure to the co-existence of hcp-T and hcp-O structure (Fig. 2b). Further increase of the $In^{3+}$ content indeed leads to the complete transition to the ccp-M structure. Such a structural transition can be predictable. With a larger τ, the weaker electron cloud extension of the cations is less likely to destroy the originally high symmetry of the ccp anion framework upon interstitial occupation of cations. Rietveld refinement of the X-ray diffraction (XRD) pattern of the $Li_3Ho_{0.6}In_{0.4}Cl_6$ sample in Fig. 2c confirmed its monoclinic structure. Meanwhile, the dramatic increase in ionic conductivity coincided with the formation of the ccp-M structure (Supplementary Fig. 2). Among the $Li_3Ho_{1-x}In_xCl_6$ samples, the highest ionic conductivity of $2.5 \times 10^{-3}\,S\,cm^{-1}$ (25 °C) was achieved by $Li_3Ho_{0.6}In_{0.4}Cl_6$. The value is more than ten times higher than that of the pristine hcp-T type $Li_3HoCl_6$. The similar structure and ionic conductivity evolution can be further confirmed in the $Li_3Ho_{1-x}Sc_xCl_6$ and $Li_3Er_{1-x}In_xCl_6$ systems (Supplementary Figs. S3−7, Supplementary Tables 8, 9).

In the case of constant Li content and unchanged $M^{3+}$ selection, the structure transition from hcp-T to ccp-M can also be realized by decreasing the ionic potential of $X^-$ with the substitution of a larger anion. Two representatives are hcp-T typed $Li_3YCl_6$[11,16] and ccp-M typed $Li_3YBr_6$[17]. When $Li^+$ and $Y^{3+}$ cations stack into the anion framework, the larger size of the $Br^-$ anion will dilute the electronic cloud

extension effect of cations, thus ensuring the high symmetry of the $Br^-$ anion framework in the ccp arrangement. However, the $Cl^-$ anion framework with a smaller anion size cannot balance the electronic cloud extension effect of cations, thus resulting in the decreased symmetry of the $Cl^-$ anion framework to the hcp arrangement. Taking hcp-T typed $Li_3YCl_6$ as a starting point, partial or total substitution of $Cl^-$ by $Br^-$ to form ccp-M typed $Li_3YCl_3Br_3$ and $Li_3YBr_6$ has been proved to be effective[17,18]. As displayed in Fig. 3a, b, the target sample of $Li_3YCl_{5.6}Br_{0.4}$ still remains the hcp-T structure, while further Br-substitution induces structural transitions to the hcp-O and finally the ccp-M phases at compositions of $Li_3YCl_{4.8}Br_{1.2}$ and $Li_3YCl_{4.5}Br_{1.5}$, respectively. Rietveld refinement of the XRD pattern reveals that the hcp-O structure of the $Li_3YCl_{4.8}Br_{1.2}$ can be indexed by the orthorhombic pnma space group (Fig. 3c, d). Moreover, the phase change from the hcp-T typed $Li_3YCl_6$ to ccp-M typed $Li_3YCl_{4.5}Br_{1.5}$ also leads to an increased ionic conductivity up to $2.16 \times 10^{-3}\,S\,cm^{-1}$ compared to $7.1 \times 10^{-5}\,S\,cm^{-1}$ of $Li_3YCl_6$ (Supplementary Fig. 8). The similar structure and ionic conductivity evolution can be further confirmed in the $Li_3ErCl_{6-x}Br_x$, $Li_3YCl_3Br_3$[17], and $Li_3YbCl_3Br_3$ halides (Supplementary Fig. S9).

We then use the cationic polarization factor to verify its feasibility in the influence on the cation stack within the same anion framework. In addition to the decreased structural symmetry from ccp to hcp anion framework due to the reduced cation polarization, further lowering the τ will boost the severe degradation of symmetry from hcp-O type ($D_{4h}$ point group, 16 symmetry elements) to hcp-T type ($D_{3d}$ point group, 12 symmetry elements) structure caused by the different cation stack within the hcp anion sublattice ($Li_3MX_6$ composition, line 1 in Fig. 4a). In addition, our previous work has proved

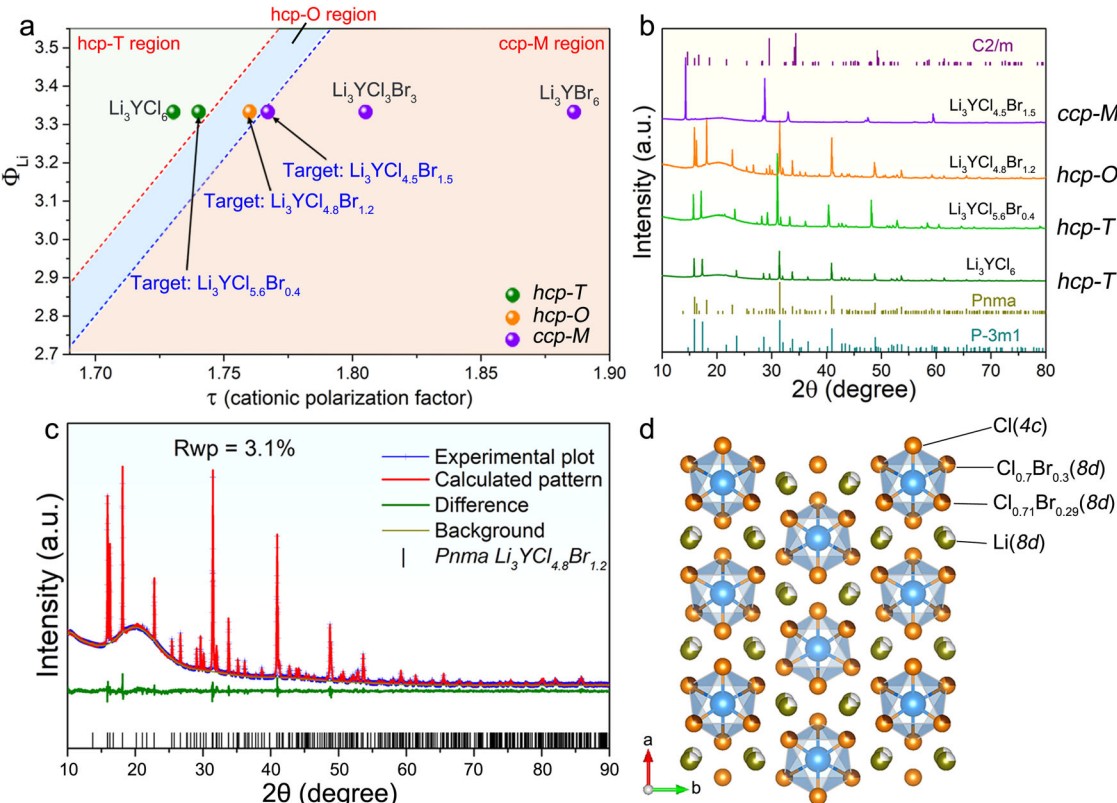

**Fig. 3 | Cationic polarization factor and its application in designing different-structured halides through the $X^-$ regulation. a** Analysis of the cationic polarization factor of $Li_3YCl_{6-x}Br_x$ halides (see Supplementary Table 10 for details). **b** XRD patterns of the targeted $Li_3YCl_{6-x}Br_x$ samples and the standard references. **c** Rietveld refinement of XRD pattern of hcp-O typed

$Li_3YCl_{4.8}Br_{1.2}$ (see Supplementary Tables 11, 12 for details). **d** Schematic illustration of the corresponding structure of hcp-O typed $Li_3YCl_{4.8}Br_{1.2}$ sample. ccp-M indicates the C2/m space group, hcp-T indicates the $P\bar{3}m1$ space group, and hcp-O indicates the Pnma space group.

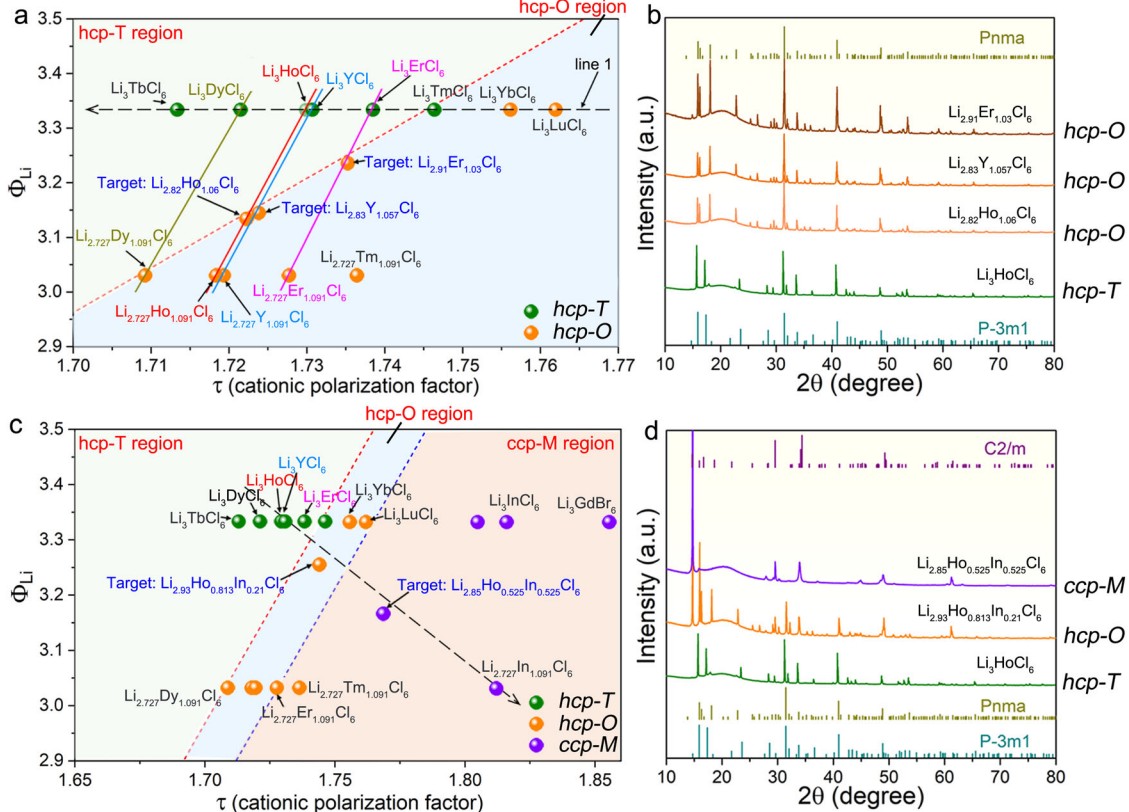

**Fig. 4 | Cationic polarization factor and its application in solving the critical points that separate hcp-T and hcp-O structured halides. a** Definition of the critical points based on the proposed cationic polarization factor of $Li_{3-3x}Dy_{1+x}Cl_6$, $Li_{3-3x}Ho_{1+x}Cl_6$, $Li_{3-3x}Y_{1+x}Cl_6$, and $Li_{3-3x}Er_{1+x}Cl_6$ halides (see Supplementary Table 13 for details). **b** XRD patterns of the targeted $Li_{2.82}Ho_{1.06}Cl_6$, $Li_{2.83}Y_{1.057}Cl_6$, and $Li_{2.91}Er_{1.03}Cl_6$ samples and the standard references. Cationic polarization factor and its application in designing different-structured halides through both lithium content and $M^{3+}$ regulation. **c** Definition of the two points based on the proposed cationic polarization factor of $Li_{3-3x}(Ho_{1-y}In_y)_{1+x}Cl_6$ halides along the line through hcp-T typed $Li_3HoCl_6$ to ccp-M typed $Li_{2.727}In_{1.091}Cl_6$ (see Supplementary Table 14 for details). **d** XRD patterns of the targeted $Li_{2.93}Ho_{0.81}In_{0.21}Cl_6$ and $Li_{2.85}Ho_{0.525}In_{0.525}Cl_6$ samples and the standard references. ccp-M indicates the C2/m space group, hcp-T indicates the P$\bar{3}$m1 space group, and hcp-O indicates the Pnma space group.

that the structural symmetry of hcp-T type $Li_3MCl_6$ (M = Dy, Ho, Y, Er, Tm) can be increased to hcp-O by changing the component to $Li_{2.727}M_{1.091}Cl_6$ though the decreased $\tau$[11]. This is reasonable due to that more vacancies are generated, which can weaken the effect of the cation stack on the basic anion framework.

By drawing the lines between $Li_3MCl_6$ and $Li_{2.727}M_{1.091}Cl_6$ halides, several intersections with different $Li_{3-3x}M_{1+x}Cl_6$ compositions can be obtained. Based on these intersections, three target compositions of $Li_{2.82}Ho_{1.06}Cl_6$, $Li_{2.83}Y_{1.057}Cl_6$, and $Li_{2.91}Er_{1.03}Cl_6$ were selected (Fig. 4a), which are close to the phase change line. The XRD patterns of these samples (Fig. 4b) demonstrate the hcp-O structure, proving the successful prediction of structural transition. The different amount of $Li^+$ and $M^{3+}$ cations that dominate the structural transition of $Li_{3-3x}M_{1+x}Cl_6$ halides is attributed to the various number of vacancies that are required to tolerate and balance the effect of the stacked cations within the anion sublattice. Compared to $Li_3ErCl_6$, the hcp-T type $Li_3HoCl_6$ with strong cation electronic cloud extension (small $\tau$) needs a large number of vacancies to weaken the cation effect to achieve higher structural symmetry (change to hcp-O type). The impedance evaluation (Supplementary Fig. 10) confirms the prediction of higher RT ionic conductivities of the hcp-O structured $Li_{2.83}Y_{1.06}Cl_6$ and $Li_{2.727}Y_{1.091}Cl_6$ ($5.7 \times 10^{-4}$ S cm$^{-1}$ and $6.9 \times 10^{-4}$ S cm$^{-1}$, respectively) than that of the hcp-T structured $Li_3YCl_6$ ($7.1 \times 10^{-5}$ S cm$^{-1}$). The improved ionic conductivities should be due to the facile diffusion in the z-direction in the orthorhombic structure and such an increase in ionic conductivity from hcp-T to hcp-O type was also displayed in the $Li_{2.727}M_{1.091}Cl_6$ (M = Dy, Ho, Y, Er, Tm) halides[11].

The structural descriptor of the cationic polarization factor was further validated in the more complex cases with simultaneous change of cation concentrations and mixing of trivalent metal cations, identifying the borderline for structural transition among the three structures (hcp-T, hcp-O, and ccp-M). Along the line between hcp-T type $Li_3HoCl_6$ and ccp-M type $Li_{2.727}In_{1.091}Cl_6$ halides, two targets of $Li_{2.93}Ho_{0.82}In_{0.21}Cl_6$ ($Li_{2.93}(Ho_{0.8}In_{0.2})_{1.023}Cl_6$) and $Li_{2.85}Ho_{0.525}In_{0.525}Cl_6$ ($Li_{2.85}(Ho_{0.5}In_{0.5})_{1.05}Cl_6$) are presented (Fig. 4c). By partial substitution of $Ho^{3+}$ with $In^{3+}$ and by the introduction of vacancies in the $Li_3HoCl_6$ base, the two target halide compositions presented the designed hcp-O and ccp-M structures as confirmed by their XRD patterns (Fig. 4d). The RT ionic conductivities along the line from $Li_3HoCl_6$ to $Li_{2.727}In_{1.091}Cl_6$ also show the gradually increasing trend from hcp-T to hcp-O and the final ccp-M structure (Supplementary Table 4).

The calculation of cationic polarization factors is highly dependent on the cation and anion radii. For $Li_aMX_b$ halides involved with $M^{4+}$ cations, such as Li-M-Zr-Cl[19], Li-M-Hf-Cl[19], it can be seen that the structure based on the cationic polarization factor calculation should be ccp-M phase (Fig. 5a, Supplementary Table 15). However, the experimental results show that these Li-M-Zr(or Hf)-Cl halides possess hcp-O structure (except Li-In-Zr-Cl structure due to the stable $Li_2ZrCl_6$ is also ccp-M structured)[20], which are different from the predicted.

The deviations are relatively interpretable. The radius difference of the cations should also be considered. The $Li_aMX_b$ halides are developed from ccp-type LiX. Due to the larger radius of halogen anions, the structure of LiX can be seen as the distribution of small $Li^+$ ions into the $X^-$ anion sublattice. When forming $Li_aMX_b$ halides, $M^{3+}$

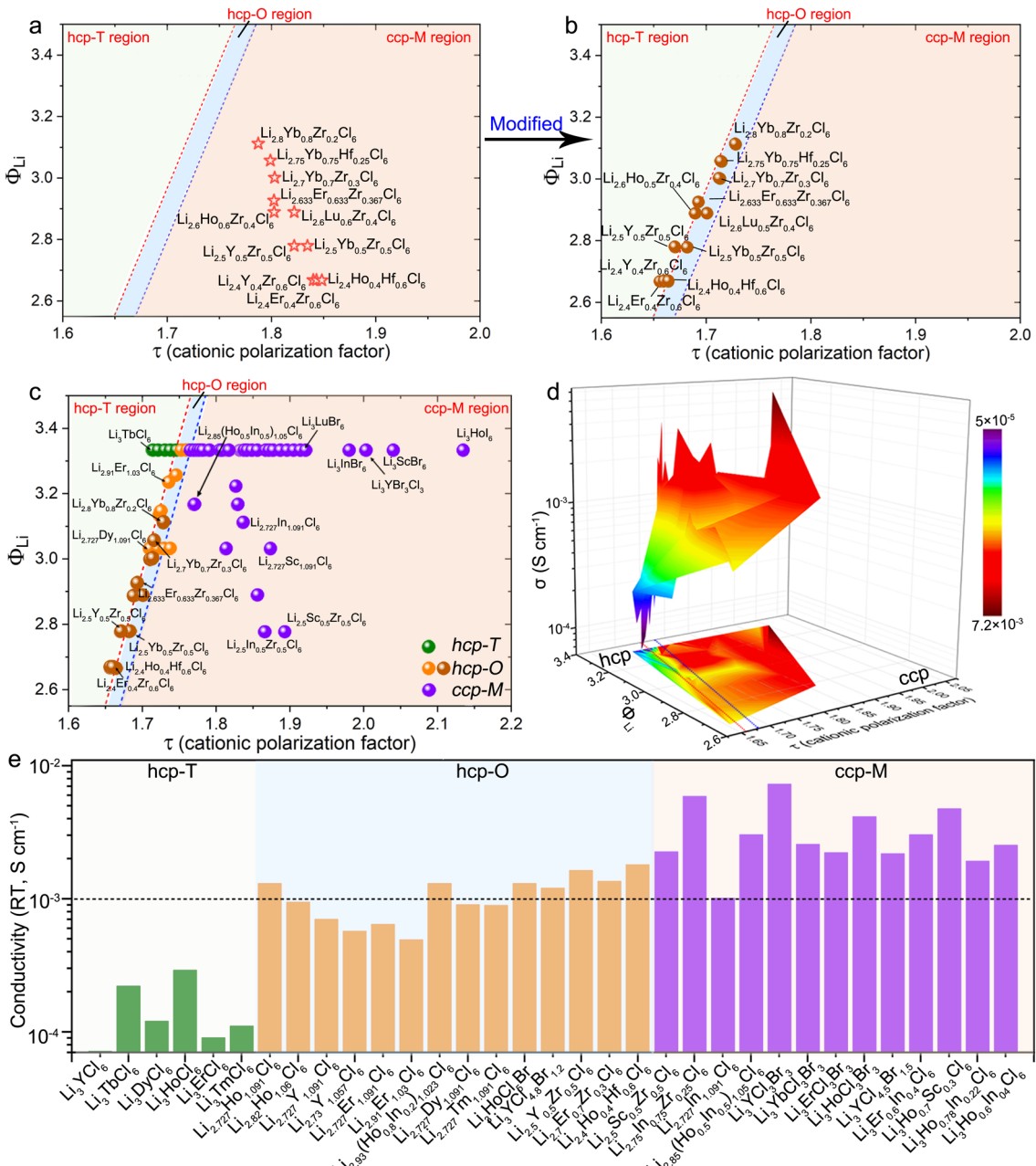

**Fig. 5 | Cation radius difference effect. a** Cationic polarization factor of the reported several Li-M-Zr-Cl halides (see Supplementary Table 15 for details). **b** Modified cationic polarization factor of the reported several Li-M-Zr-Cl halides (see Supplementary Table 15 for details) considering the cation radius difference. **c** The cationic polarization factor of $Li_aMX_b$ halides that reported and developed in

this study (see Supplementary Table 4 for details). **d** Topography of the RT ionic conductivities of $Li_aMX_b$ halides. **e** Comparison of the RT ionic conductivities of the halide SSEs based on the proposed τ. ccp-M indicates the C2/m space group, hcp-T indicates the P$\bar{3}$m1 space group, and hcp-O indicates the Pnma space group.

cations are further added to the system. The highly symmetrical ccp structure can be maintained if there's a minor difference in the cation radius. Thus, in the case of the same anion, $Li_3InCl_6$, and $Li_3ScCl_6$ with the smallest cation difference can keep the ccp structure (high symmetry). While $Li_3YbCl_6$ and $Li_3LuCl_6$ with moderate differences lead to decreased symmetry to hcp-O structure (moderate symmetry) and other $Li_3MCl_6$ halides (M = Y, Tb-Tm) with the largest difference result in a further decrease of symmetry to hcp-T structure (Supplementary Table 16). This further explains why $Li_{3-x}In_{1-x}Zr_xCl_6$ can achieve ccp structure while $Li_{3-x}Yb_{1-x}Hf_xCl_6$, $Li_{3-x}Yb_{1-x}Zr_xCl_6$, $Li_{3-x}Y_{1-x}Zr_xCl_6$, and $Li_{3-x}Er_{1-x}Zr_xCl_6$ halides possess the hcp-O structure (Supplementary Table 17). Herein, the cationic polarization factor τ is modified for

these $Li_{3-x}M_{1-x}^{3+}M_x^{4+}X_6$ halides with a large cation radius difference.

$$\tau = \frac{6-x}{6} \frac{\Sigma \Phi cations}{\Sigma \Phi X} = \frac{6-x}{6} \frac{\Phi Li + \Sigma \Phi M}{\Sigma \phi X} \quad (2)$$

where the $x$ is the molar content of tetravalent cations. After modeling modification, the optimized τ values of these $Li_{3-x}M_{1-x}^{3+}M_x^{4+}X_6$ halides are shown in Fig. 5b. We have further added several typical examples involving $Zr^{4+}$ and $Hf^{4+}$ as shown in Supplementary Table 18 and Supplementary Fig. 11. It can be seen that most of these halides can still locate in the hcp-O region. Thus, the whole phase map was further presented in Fig. 5c, which also includes the typical clarified samples in

this work. Furthermore, the topography of the RT ionic conductivities of $Li_aMX_b$ halides (Fig. 5d, e) presents that most reported halides with conductivity over $10^{-3}$ S cm$^{-1}$ are located in the ccp anion sublattice region. In contrast, halides with hcp structure possess relatively low conductivity.

Overall, most of the $Li_aMX_b$ halides are derived from basic LiCl. When introducing different multivalent cations into the LiCl ccp structure, it can be regarded as putting charged balls with varied radii into this system. The proposed $\tau$ here describes the ability of the material to maintain the basic ccp skeleton, which involves three key factors. The first one is the size difference among the cations (Li$^+$ and M$^{x+}$) and the increase of size difference will destroy the high symmetry from ccp to hcp-O and finally to the hcp-T structure with the lowest symmetry. The second one is the anion framework. Normally, a larger-sized anion framework can reduce the cation effect, such as the ccp-structured $Li_3YBr_6$ compared with hcp-structured $Li_3YCl_6$[21]. The third one is introducing more vacancy to weaken the cation effect, such as the hcp-O structured $Li_{3-3x}M_{1+x}Cl_6$ compared with hcp-T structured $Li_3MCl_6$[11].

Note that some other halide-based SSEs with different structures, such as nine-coordinated LaCl$_3$-based SSEs are reported lately[3]. The formation of this structure is the result that f electrons also participate in the bonding of more Cl$^-$. Thus, a more covalent composition of the chemical bonds is involved, which is not the scope of the radius ratio and proposed $\tau$ here that mainly focuses on ionic bonds. Other glassy halides[2] that recently reported are also not discussed here due to their glassy structure.

## Discussion

In summary, we mapped the structural landscape of the existing and potential superionic Li$^+$ conducting halide SSEs by proposing a proposed cationic polarization factor, $\tau$. The main difference among the hcp-T, hcp-O, and ccp-M structures of halide SSEs is the polarization among the cationic and anionic sublattices. The $\tau$ can not only successfully classify the experimentally observed halide structures but also predict the structure of the unreported halide SSE compositions. Guided by the $\tau$, more than 10 lithium halide SSEs with RT ionic conductivities over $10^{-3}$ S cm$^{-1}$ have been identified and synthesized. Moreover, the $\tau$ can be used to predict the phase transition compositions of lithium halide SSEs with different cation concentrations and different mixtures of multivalent cations. Well-designed phase transition can effectively tune the Li$^+$-conducting behavior of halide SSEs. In most cases, highly improved ionic conductivity can be achieved by tuning the base structure from hcp to ccp. The deficiency of the $\tau$ for some halide electrolytes with tetravalent cations arises mainly from the large cation radius difference, which leads to the decrease of symmetry and thus influences the structure. Due to the simplicity and accuracy of the proposed $\tau$, we expect its use to accelerate the exploration and design of lithium halide superionic conductors.

From the viewpoint of crystal structures, the halide SSEs discussed in our work are mainly based on octahedral coordination similar to the studies of various tolerance factors proposed for perovskite materials. Moreover, the cationic polarization factor proposed here is based on the ion stacking in ionic crystals. For other classes of ion conductors (such as sulfides, and polyanionic oxides), most of these materials contain structural frameworks that involve covalent bonds, such as $PS_4^{3-}$, $PO_4^{3-}$, and $SiO_4^{4-}$. The crystal structures of these ion conductors possess various coordination conditions (four-coordination for $PS_4^{3-}$, eight-coordinated LaO$_8^{13-}$, etc.), which are totally different from that of halide SSE discussed here. Thus, the cationic polarization factor is not suitable for these materials. Precisely because of the structural and chemical distinctions, the Li$^+$ migration through these frameworks will lead to the different phenomena and diffusion mechanisms in halide SSEs. For ion conductors with specific crystal structures, it's still possible to investigate the formability by similar rules. We also hope that this work will be helpful to the modification of the tolerance factor for perovskite materials by further considering ionic charges rather than just the ionic radii effect.

## Methods
### Sample synthesis
The resulting materials were prepared by a co-melt and recrystallization method. The starting materials include lithium chloride (LiCl, Alfa Aesar, 99.9%), lithium bromide (LiBr, Alfa Aesar, 99.9%), trivalent rare earth bromides (MBr$_3$, Alfa Aesar, >99.9%), and metal chlorides (MCl$_3$, MCl$_4$, Alfa Aesar, > 99.9%). Raw materials with stoichiometric ratio were directly put and sealed in a quartz tube at -10 Pa under vacuum. The quartz tube was heated for 4 h to reach 550–650 °C and kept at 650 °C for 12 h. Then the quartz tube was cooled down to 25 °C within 10–48 h.

### Characterizations
Powder X-ray diffraction (XRD) patterns were characterized on a Bruker AXS D8 Advance (Cu Kα radiation, λ = 1.54178 Å). The samples were loaded into a homemade air-tight holder to avoid air exposure.

### Conductivity measurements
All the alternating current (AC) impedance was measured using Bio-Logic VMP3 electrochemical working station with an amplitude of 10 mV at frequencies from 7 MHz to 1 Hz. The pressing and cell assembling operations were conducted in the Ar-filled glove box (O$_2$ < 1 ppm, H$_2$O < 1 ppm). The synthesized solid electrolyte powder was pressed into pellets (10 mm diameter, thickness around 10 mm) at -380 MPa and attached with stainless steel rod electrodes. An external pressure of -200 Mpa was added during the impedance tests.

## Data availability
The data of this study are available from the corresponding authors upon request. Source data are provided with this paper.

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

## Acknowledgements

This research was supported by the National Natural Science Foundation of China (No. 22379127 [J.W.L.], 52176185[T.M.]), the Guangdong High-level Innovation Institute project (2021B0909050001 [J.W.L.]), the Natural Sciences and Engineering Research Council of Canada (NSERC), Canada Research Chair Program (CRC), Canada Foundation for Innovation (CFI), Ontario Research Fund, the University of Western Ontario, and Eastern Institute of Techonology, Ningbo.

## Author contributions

X.L., J.W.L., and X.S. conceived the project. X.L., J.K., C.Z., Y.X., T.M., and R.L. performed the electrolyte synthesis and electrochemical performance characterizations. X.L., J.L., and J.K. interpreted the data and wrote the manuscript. J.W.L. and X.S. supervised the project. All the authors participated in the reviewing and editing of the manuscript.

## Competing interests

The authors declare no competing interests.
