## [Peer Review File · Nature Communications]

REVIEWER COMMENTS

Reviewer #1 (Remarks to the Author):

In this manuscript, the authors propose a polarization factor which can be utilized for the design of halide solid electrolyte Li_3MCl_6 . While this work is intriguing, it does not meet the high-quality standards expected by Nature Communications. Before publishing, it is essential to tackle the following issues:

1) The phase transition of Li_3MCl_6 (among CCP-M, hcp-O, and hcp-T) and the high ionic conductivity of the monoclinic phase have been widely reported in the literature. [Science Advances, 2022, 8(36), eadc9516; ACS Energy Letters, 2022, 7(5), 1776-1805.] The authors should clearly provide the novelty of their work.

2) The authors demonstrated a general design rule for Li_3MCl_6 , but they only presented a limited number of new Li_3MCl_6 . The reviewer concerns regarding the applicability of the structural design rule for Li_3MCl_6 , suggesting that it may be limited to a specific region and may not offer new hypothetical Li_3MCl_6 structures. Please provide various new hypothetical Li_3MCl_6 structures that can be synthesized by other researchers.

3) The structural design rule was modified for Zr^{4+} , but the reviewer concerns regarding the effectiveness of the design rule for M^{4+}

4) This study includes a limited range of Li_3MCl_6 , with the elements primarily restricted to the costly lanthanides. Furthermore, the comprehensive examination of the effect of Br has not been conducted.

Reviewer #2 (Remarks to the Author):

The present work deals with the systematic investigation of novel compositions of halide-based Li-ion conductors. Using a cationic polarization factor, the authors reveal compositions with increased

ionic conductivity. The work presents a novel approach for the design and improvement of halide ion conductors and can be accepted for publication after minor revision. A point-by-point list of remarks is given in the following.

- At the beginning of page 2, there is an error in the citing literature. [...]different Li⁺ migration properties. Please correct this.
- In Figure 1d the crystallographic axis should be included to get a better understanding of the structural changes
- As the cationic polarization factor is relatively new in the field of solid electrolytes, can the authors extend the explanation of how it was calculated (see page 4)
- What does “weighted ionic potential of lithium ions” mean? Please explain this in more detail.
- Please add a paragraph into the conclusion section discussing the possibility to transfer such concept of the cationic polarization factor to other classes of ion conductors.

Reviewer #3 (Remarks to the Author):

Halides are currently a hot topic for SEs. In this manuscript, the authors try to give and prove the structural regulation and evolution of halide SEs by using the cationic polarization factor, etc. They provide a predicted model of Li-M-X phase diagram, which is highly related to the Li-ion transport capability. This work is important for the halide design and may be a good reference for other systems. It is also interesting that they provide over 10 Li-halide electrolytes possessing high conductivities over 1 mS/cm. I think the prediction is reasonable. Therefore, I recommend to publish the manuscript in Nature Communication.

Comments:

1. Why do you use Shannon crystal radii rather than the Pauling radius of ions, as we know Pauling radius is also widely used? What are the differences for them and the scope of their respective application? I suggest to give more detailed information in the manuscript.
2. The structure model is based on the factors, e.g., ion polarization. I wonder how much influence the electronic structure of different elements has on this model?
3. The phase structures of hcp-T, hcp-O, and ccp-M show different ion conduction rates. Why and what are the rules about them? Does it mean that the symmetry of the halide structure and/or the arrangement of the anion framework have a significant influence on ion transport?

4. The oxychloride-based and UCl_3 -type SEs e.g., Li-La-Ta-Cl have been reported recently, how about to use this phase diagram model in these novel electrolytes? Have you considered to use the model in other systems say, sulfide SEs?

5. The references for the definition of ionic potential should be cited. Regarding the formula $\Phi_{Li} = n_{Li} \times \varphi_{Li}$, φ_{Li} is not defined.

Response to the Comments of Reviewers' On "Structural Regulation of Halide Superionic Conductors for All-Solid-State Lithium Batteries" (Research article, *Nature Communications*, NCOMMS-23-26249-T)

We highly appreciate the reviewers' recommendation on publishing this paper in *Nature Communications*. We are also very grateful to the reviewers for providing constructive comments and insightful suggestions for further improving the quality of the manuscript. The manuscript has been revised with great effort. We carefully addressed the concerns and corrected any errors. The specific responses and revisions are listed below. Many thanks!

The most significant improvements in the revised version of the manuscript can be summarized as follows:

- 1) We have added provided more Li_aMX_b halides that follow the cationic polarization rules.
- 2) We further proved the effectiveness of the design rule for M^{4+} .
- 3) We further conducted a comprehensive examination of the effect of Br.
- 4) We have discussed the possibility of transferring the concept of the cationic polarization factor to other classes of ion conductors.
- 5) Other concerns have also been addressed.

REVIEWER REPORTS:

Referee: #1

Comments to the Author

In this manuscript, the authors propose a polarization factor which can be utilized for the design of halide solid electrolyte Li_3MCl_6 . While this work is intriguing, it does not meet the high-quality standards expected by Nature Communications. Before publishing, it is essential to tackle the following issues:

Response: Many thanks for your helpful comments.

(1) The phase transition of Li_3MCl_6 (among CCP-M, hcp-O, and hcp-T) and the high ionic conductivity of the monoclinic phase have been widely reported in the literature. [Science Advances, 2022, 8(36), eadc9516; ACS Energy Letters, 2022, 7(5), 1776-1805.] The authors should clearly provide the novelty of their work.

Response: Many thanks for the suggestions. We acknowledge the previous works about the phase transition of Li_3MCl_6 and the high ionic conductivity of the monoclinic phase, whereas we hope to provide a principle for the rational design of halide SSEs and the determination of the phase transition of halide SSEs when regulating the composition, which further induces changed Li^+ -conducting behavior. We highlight the novelties and importance of this work in the discussion below.

The development of halide SSEs has been very limited compared to other types of well-developed solid-state electrolytes (SSEs) such as oxides, polymers, borohydrides, and sulfides. Until now, there are many successful examples of halide SSEs that can achieve room-temperature (RT) ionic conductivity over $10^{-3} \text{ S cm}^{-1}$, i.e., Li_3YBr_6 , Li_3InCl_6 , and zirconium substituted $\text{Li}_{3-x}\text{M}_{1-x}\text{Zr}_x\text{Cl}_6$. How the composition determines the structural chemistry is crucial for the ionic conductivity of halide SSEs but very challenging to predict. Therefore, it is essential and urgent to make it possible to predict the stacking structures and conductive properties of halide SSEs by capturing the key interactions and other factors.

For the concern of the related previous reports, firstly, these two works are review papers and only limited sections that discuss the structure and conduction behavior of halide SSEs. Secondly, these two works only roughly mentioned the different cations/anions radii and the ratio of ionic radii of cation M and anion X ($r_{M/X}$) effects on the final halide SSEs. Thirdly, the radii effect only accounts for the difference in cation/anion radii, which makes it impossible to predict the structure of halides SSEs with the same metal cations and anions (for example, *hcp-T* typed Li_3HoCl_6 and *hcp-O* typed $\text{Li}_{2.727}\text{Ho}_{1.091}\text{Cl}_6$, *hcp-T* typed Li_3ErCl_6 and *hcp-O* typed $\text{Li}_{2.727}\text{Er}_{1.091}\text{Cl}_6$), let alone give further guidance of structure regulation.

In our work, we introduce the “cationic polarization factor” which provides an estimate of the phase transition among the different structures of halide SSEs, which further induces different Li-conducting behavior. By supplementing this principle with rational design and preparation of more than 10 lithium halide electrolytes with high conductivity over $10^{-3} \text{ S cm}^{-1}$, we establish that there should be a variety of promising halide-based SSEs that have yet to be discovered and developed. ***The key novelty and advancements of our study*** are that, for the first time, we proposed a cationic polarization factor, which captures the key role of cation/anion interactions on the final stacking structures that can chart the landscape of the existing and future lithium halide SSEs. The cationic polarization factor enables the classification of experimentally observed and constitutes the prediction of those still unexplored lithium halide SSEs. Guided by the cationic polarization factor, more than 10 lithium halide SSEs with RT ionic conductivities over $10^{-3} \text{ S cm}^{-1}$ have been identified and synthesized. The cationic polarization factor further allows the determination of the phase transition of halide SSEs when regulating the composition, which further induces changed Li^+ -conducting behavior.

(2) The authors demonstrated a general design rule for Li_3MCl_6 , but they only presented a limited number of new Li_3MCl_6 . The reviewer concerns regarding the applicability of the structural design rule for Li_3MCl_6 , suggesting that it may be limited to a specific region and may not offer new hypothetical Li_3MCl_6 structures. Please provide various new hypothetical Li_3MCl_6 structures that can be synthesized by other researchers.

Response: Many thanks for the suggestions. As shown in **Figure R1**, we have proposed four strategies and related examples that follow the structural design rule for Li_3MCl_6 , including cation

regulation, anion regulation, cation concentration regulation, and combined regulation. Based on the proposed routes, we have synthesized more than twenty Li_3MX_6 halides that follow the structural design rule (Table R1).

Figure R1. Four strategies for the regulation of structures and conductivities of Li_3MX_6 halides.

Table R1. Calculated molar content proportionally ionic potential of different ions and corresponding cationic polarization factors for Li_aMX_b halides. The reported conductivities of Li_aMX_b halides are also presented.

Samples	Halides	Structure	$\tau = (\Phi\text{Li} + \Sigma\Phi\text{M})/\Sigma\Phi\text{X}$	$\phi\text{Li} = n_{\text{Li}} \cdot I_{\text{Li}}$ (\AA^{-1})	$\phi\text{M} = n_{\text{M}} \cdot I_{\text{M}}$ (\AA^{-1})	$\phi\text{X} = n_{\text{X}} \cdot I_{\text{X}}$ (\AA^{-1})	Conductivity (S cm^{-1} , 25 °C)
1	$\text{Li}_3\text{Er}_{0.95}\text{In}_{0.05}\text{Cl}_6$	hcp-T	1.7423	3.3333	2.9265	3.5928	3.3×10^{-4}
2	$\text{Li}_{2.82}\text{Ho}_{1.06}\text{Cl}_6$	hcp-O	1.7223	3.1333	3.0547	3.5928	9.4×10^{-4}
3	$\text{Li}_{2.83}\text{Y}_{1.057}\text{Cl}_6$	hcp-O	1.7238	3.1444	3.049	3.5928	5.7×10^{-4}
4	$\text{Li}_{2.911}\text{Er}_{1.03}\text{Cl}_6$	hcp-O	1.7353	3.2344	3	3.5928	4.9×10^{-4}
5	$\text{Li}_{2.93}(\text{Ho}_{0.8}\text{In}_{0.2})_{1.023}\text{Cl}_6$	hcp-O	1.7448	3.2555	3.0131	3.5928	1.3×10^{-3}
6	$\text{Li}_3\text{HoCl}_5\text{Br}$	hcp-O	1.7539	3.3333	2.8818	3.5435	1.3×10^{-3}
7	$\text{Li}_3\text{ErCl}_5\text{Br}$	hcp-O	1.7626	3.3333	2.9126	3.5435	1.14×10^{-3}
8	$\text{Li}_3\text{YCl}_{4.8}\text{Br}_{1.2}$	hcp-O	1.7597	3.3333	2.8846	3.5336	1.2×10^{-3}
9	$\text{Li}_{2.727}\text{In}_{1.091}\text{Cl}_6$	ccp-M	1.8125	3.0300	3.4819	3.5928	1×10^{-3}
10	$\text{Li}_{2.85}(\text{Ho}_{0.5}\text{In}_{0.5})_{1.05}\text{Cl}_6$	ccp-M	1.7688	3.1666	3.1884	3.5928	3.1×10^{-3}
11	$\text{Li}_3\text{HoCl}_3\text{Br}_3$	ccp-M	1.8042	3.3333	2.8818	3.4449	4.1×10^{-3}
12	$\text{Li}_3\text{HoCl}_4\text{Br}_2$	ccp-M	1.7787	3.3333	2.8818	3.4942	2.3×10^{-3}
13	$\text{Li}_3\text{ErCl}_3\text{Br}_3$	ccp-M	1.8131	3.3333	2.9126	3.4449	2.2×10^{-3}
14	$\text{Li}_3\text{ErCl}_4\text{Br}_2$	ccp-M	1.7875	3.3333	2.9126	3.4942	2.0×10^{-3}
15	$\text{Li}_3\text{YbCl}_3\text{Br}_3$	ccp-M	1.8316	3.3333	2.9762	3.4449	2.3×10^{-3}
16	$\text{Li}_3\text{YbCl}_4\text{Br}_2$	ccp-M	1.8057	3.3333	2.9762	3.4942	2.0×10^{-3}
17	$\text{Li}_3\text{YCl}_{4.5}\text{Br}_{1.5}$	ccp-M	1.7670	3.3333	2.8846	3.5189	2.1×10^{-3}
18	$\text{Li}_3\text{Er}_{0.76}\text{In}_{0.24}\text{Cl}_6$	hcp-O+ ccp-M	1.7571	3.3333	2.9795	3.5928	2.9×10^{-3}
19	$\text{Li}_3\text{Er}_{0.6}\text{In}_{0.4}\text{Cl}_6$	ccp-M	1.7696	3.3333	3.0242	3.5928	3×10^{-3}
20	$\text{Li}_3\text{Ho}_{0.7}\text{Sc}_{0.3}\text{Cl}_6$	ccp-M	1.7723	3.3333	3.0342	3.5928	4.7×10^{-3}
21	$\text{Li}_3\text{Ho}_{0.78}\text{In}_{0.22}\text{Cl}_6$	hcp-O+ ccp-M	1.7488	3.3333	2.9499	3.5928	1.9×10^{-3}
22	$\text{Li}_3\text{Ho}_{0.68}\text{In}_{0.32}\text{Cl}_6$	ccp-M	1.7575	3.3333	2.9809	3.5928	2.3×10^{-3}
23	$\text{Li}_3\text{Ho}_{0.6}\text{In}_{0.4}\text{Cl}_6$	ccp-M	1.7644	3.3333	3.0057	3.5928	2.5×10^{-3}

We have further added several typical $\text{Li}_{3-x}\text{M}_{1-x}^{3+}\text{M}_x^{4+}\text{X}_6$ halides that contain M^{4+} elements as presented in Table R2 and Figure R2.

Table R2. Calculated molar content proportionally ionic potential of different ions and corresponding cationic polarization factors for $\text{Li}_{3-x}\text{M}_{1-x}^{3+}\text{M}_x^{4+}\text{X}_6$ halides.

Sample No.	Halides	Structure	$\tau = (\phi\text{Li} + \Sigma\phi\text{M}) / \Sigma\phi\text{X}$	Modified $\tau = \frac{6-x}{6}(\phi\text{Li} + \Sigma\phi\text{M}) / \Sigma\phi\text{X}$	$\phi\text{Li} = n_{\text{Li}} \cdot I_{\text{Li}} (\text{\AA}^{-1})$	$\phi\text{M} = n_{\text{M}} \cdot I_{\text{M}} (\text{\AA}^{-1})$	$\phi\text{X} = n_{\text{X}} \cdot I_{\text{X}} (\text{\AA}^{-1})$	Conductivity (S cm^{-1} , 25°C)	Ref.
1	$\text{Li}_{2.5}\text{Yb}_{0.5}\text{Zr}_{0.5}\text{Cl}_6$	hcp-O	1.8346	1.6818	2.7778	3.8137	3.5928	1.63×10^{-3}	This work
2	$\text{Li}_{2.6}\text{Y}_{0.6}\text{Hf}_{0.4}\text{Cl}_6$	hcp-O	1.8097	1.6891	2.8889	3.6131	3.5928	1.31×10^{-3}	This work
3	$\text{Li}_{2.7}\text{Er}_{0.7}\text{Zr}_{0.3}\text{Cl}_6$	hcp-O	1.7909	1.7104	3.0000	3.4342	3.5928	1.35×10^{-3}	This work
4	$\text{Li}_{2.4}\text{Ho}_{0.4}\text{Hf}_{0.6}\text{Cl}_6$	hcp-O	1.8489	1.6640	2.6666	3.9763	3.5928	6.81×10^{-4}	This work

Figure R2. (a) XRD pattern and (b) EIS spectra of $\text{Li}_{2.6}\text{Y}_{0.6}\text{Hf}_{0.4}\text{Cl}_6$ (thickness of 0.7 mm), (c) XRD pattern and (d) EIS spectra of $\text{Li}_{2.4}\text{Ho}_{0.4}\text{Hf}_{0.6}\text{Cl}_6$ (thickness of 0.8 mm), (e) XRD pattern and (f) EIS spectra of $\text{Li}_{2.5}\text{Yb}_{0.5}\text{Zr}_{0.5}\text{Cl}_6$ (thickness of 1.2 mm), (g) XRD pattern and (h) EIS spectra of $\text{Li}_{2.7}\text{Er}_{0.7}\text{Zr}_{0.3}\text{Cl}_6$ (thickness of 0.8 mm).

Thus, we synthesized at least 27 halide SSEs, including 23 Li_3MX_6 compounds and 4 typical $\text{Li}_{3-x}\text{M}_{1-x}^{3+}\text{M}_x^{4+}\text{X}_6$ compounds. All the synthesized materials are based on the proposed design rule for anion-sublattice-based halide SSEs. It is believed that the proposed design principles can be broadly applied to the synthesis of a wide variety of halide SSEs with high ionic conductivities.

We have further revised in the manuscript and supporting information and highlighted.

(3) The structural design rule was modified for Zr^{4+} , but the reviewer concerns regarding the effectiveness of the design rule for M^{4+} .

Response: Many thanks for the suggestions.

The structural design rule in this work mainly focuses on Li_aMX_b halides that developed from *ccp*-type LiX . The structure of LiX can be seen as the distribution of small Li^+ ions into the X^- anion sublattice. Different types of M^{3+} cations are further added to the system to form Li_aMX_b halides. The highly symmetrical *ccp* structure can be maintained if there's a minor difference in the cation radius (such as Li_3InCl_6 , and Li_3ScCl_6 with the highest symmetry). Whereas Li_3YbCl_6 and Li_3LuCl_6 with moderate differences lead to decreased symmetry to *hcp-O* structure (moderate symmetry) and other Li_3MCl_6 halides ($M = Y, Tb-Tm$) with the largest difference result in a further decrease of symmetry to *hcp-T* structure.

Thus, for the $Li_{3-x}M_{1-x}^{3+}M_x^{4+}X_6$ halides with a small cation radius difference (Table R3), their structures fit the structural design rule well without modification, examples include $Li_{3-x}In_{1-x}Zr_xCl_6^{1-3}$, $Li_{3-x}Sc_{1-x}Zr_xCl_6^3$ and $Li_{3-x}In_{1-x}Hf_xCl_6^4$ halides (Figure R3,4).

Table R3. Cation radius differences in $Li_{3-x}M_{1-x}^{3+}M_x^{4+}X_6$ halides.

Halides		Cation radius difference between M^{3+} and Li^+ (Å)	Cation radius difference between M^{4+} and Li^+ (Å)	Cation radius difference between M^{3+} and M^{4+} (Å)
Small cation radius differences	$Li_{3-x}In_{1-x}Zr_xCl_6$	0.04	-0.04	0.08
	$Li_{3-x}Sc_{1-x}Zr_xCl_6$	-0.015	-0.04	0.0025
	$Li_{3-x}In_{1-x}Hf_xCl_6$	0.04	-0.05	0.09
Large cation radius differences	$Li_{2.8}Yb_{0.8}Zr_{0.2}Cl_6$	0.108	-0.04	0.148
	$Li_{2.75}Yb_{0.75}Hf_{0.25}Cl_6$ (500 °C)	0.108	-0.05	0.158
	$Li_{2.7}Yb_{0.7}Zr_{0.3}Cl_6$	0.108	-0.04	0.148
	$Li_{2.633}Er_{0.633}Zr_{0.367}Cl_6$	0.13	-0.04	0.17
	$Li_{2.5}Yb_{0.5}Zr_{0.5}Cl_6$	0.108	-0.04	0.148
	$Li_{2.5}Y_{0.5}Zr_{0.5}Cl_6$	0.14	-0.04	0.18
	$Li_{2.4}Er_{0.4}Zr_{0.6}Cl_6$	0.13	-0.04	0.17
	$Li_{2.4}Y_{0.4}Zr_{0.6}Cl_6$	0.14	-0.04	0.18
$Li_{2.6}Lu_{0.6}Zr_{0.4}Cl_6$	0.101	-0.04	0.141	

$\text{Li}_{2.6}\text{Ho}_{0.6}\text{Zr}_{0.4}\text{Cl}_6$	0.141	-0.04	0.181
--	--------------	-------	--------------

Figure R3. The cationic polarization factors of $\text{Li}_{3-x}\text{In}_{1-x}\text{Zr}_x\text{Cl}_6$ and $\text{Li}_{3-x}\text{Sc}_{1-x}\text{Zr}_x\text{Cl}_6$ halides.¹⁻³

Figure R4. The cationic polarization factors of $\text{Li}_{3-x}\text{In}_{1-x}\text{Hf}_x\text{Cl}_6$ halides.⁴

As presented in **Table R3**, for these $\text{Li}_{3-x}\text{M}_{1-x}^{3+}\text{M}_x^{4+}\text{X}_6$ halides with a large cation radius difference, a coefficient was introduced for the calculation of the cationic polarization factor τ for in our work. Examples include $\text{Li}_{3-x}\text{Yb}_{1-x}\text{Zr}_x\text{Cl}_6$ ^{5, 6}, $\text{Li}_{3-x}\text{Yb}_{1-x}\text{Hf}_x\text{Cl}_6$ ⁶, $\text{Li}_{3-x}\text{Y}_{1-x}\text{Zr}_x\text{Cl}_6$ ⁷, $\text{Li}_{3-x}\text{Er}_{1-x}\text{Zr}_x\text{Cl}_6$ ^{7, 8}, $\text{Li}_{3-x}\text{Ho}_{1-x}\text{Zr}_x\text{Cl}_6$ ⁹, $\text{Li}_{3-x}\text{Lu}_{1-x}\text{Zr}_x\text{Cl}_6$ ⁹, etc. We have further added several typical examples involving Zr^{4+} and Hf^{4+} as shown in **Table R4** and **Figure R5**. Moreover, we found that it's hard to synthesize Li_aMX_b halides with other M^{4+} chemicals since most other M^{4+} chemicals such as TiCl_4 , SiCl_4 , GeCl_4 , and SnCl_4 are liquids.

Table R4. Calculated molar content proportionally ionic potential of different ions and corresponding cationic polarization factors (before and after modification) for $\text{Li}_{3-x}\text{M}_{1-x}^{3+}\text{M}_x^{4+}\text{X}_6$ halides.

Sample No.	Halides	Structure	$\tau = (\phi\text{Li} + \Sigma\phi\text{M}) / \Sigma\phi\text{X}$	Modified $\tau = \frac{6-x}{6}(\phi\text{Li} + \Sigma\phi\text{M}) / \Sigma\phi\text{X}$	$\phi\text{Li} = n_{\text{Li}} \cdot I_{\text{Li}} (\text{\AA}^{-1})$	$\phi\text{M} = n_{\text{M}} \cdot I_{\text{M}} (\text{\AA}^{-1})$	$\phi\text{X} = n_{\text{X}} \cdot I_{\text{X}} (\text{\AA}^{-1})$	Conductivity (S cm^{-1} , 25 °C)	Ref.
1	$\text{Li}_{2.5}\text{Yb}_{0.5}\text{Zr}_{0.5}\text{Cl}_6$	hcp-O	1.8346	1.6818	2.7778	3.8137	3.5928	1.63×10^{-3}	This work
2	$\text{Li}_{2.6}\text{Y}_{0.6}\text{Hf}_{0.4}\text{Cl}_6$	hcp-O	1.8097	1.6891	2.8889	3.6131	3.5928	1.31×10^{-3}	This work
3	$\text{Li}_{2.7}\text{Er}_{0.7}\text{Zr}_{0.3}\text{Cl}_6$	hcp-O	1.7909	1.7104	3.0000	3.4342	3.5928	1.35×10^{-3}	This work
4	$\text{Li}_{2.4}\text{Ho}_{0.4}\text{Hf}_{0.6}\text{Cl}_6$	hcp-O	1.8489	1.6640	2.6666	3.9763	3.5928	6.81×10^{-4}	This work

Figure R5. (a) XRD pattern and (b) EIS spectra of $\text{Li}_{2.6}\text{Y}_{0.6}\text{Hf}_{0.4}\text{Cl}_6$ (thickness of 0.7 mm), (c) XRD pattern and (d) EIS spectra of $\text{Li}_{2.4}\text{Ho}_{0.4}\text{Hf}_{0.6}\text{Cl}_6$ (thickness of 0.8 mm), (e) XRD pattern and (f) EIS spectra of $\text{Li}_{2.5}\text{Yb}_{0.5}\text{Zr}_{0.5}\text{Cl}_6$ (thickness of 1.2 mm), (g) XRD pattern and (h) EIS spectra of $\text{Li}_{2.7}\text{Er}_{0.7}\text{Zr}_{0.3}\text{Cl}_6$ (thickness of 0.8 mm).

We further added these tables and figures in the revised manuscript and supporting information and highlighted.

(4) This study includes a limited range of Li_3MCl_6 , with the elements primarily restricted to the costly lanthanides. Furthermore, the comprehensive examination of the effect of Br has not been conducted.

Response: Many thanks for the suggestions. The price and corresponding abundance in Earth's crust of different candidate elements are shown in **Figure R6**. It's apparent that the cheapest candidates include Al, La, Ce, and Zr.

Figure R6. The abundance in Earth's crust and the corresponding price of different candidate elements.

For Al, the formation of Li_aAlX_b follows the basic rules of ion stacking and can be divided simply by the cation/anion radius ratio as we presented in Figure 1e and **Table R5**. The Al^{3+} in Li_3AlF_6 is six-coordinated while four-coordinated in LiAlCl_4 and LiAlBr_4 .

Table R5. The summary of the cation, halogen anion crystal radius, and their ratio of the typical Li-Al-X halides.

Halides	cation Radius (Å)	Halogen anion Radius (Å)	radius ratio of X-
Li_3AlF_6	0.675	1.19	0.5672
LiAlCl_4	0.53	1.67	0.31737
LiAlBr_4	0.53	1.82	0.2912

For La and Ce, their cation radii are too large to stack into the LiX structure (La^{3+} , 1.172 Å for six-coordinated compounds and 1.356 for nine-coordinated compounds; Ce^{3+} , 1.15 Å for six-coordinated compounds and 1.336 for nine-coordinated compounds). There's no solid-state Li_aLaX_b or Li_aCeX_b that can be synthesized from the LiX-LaX_3 or LiX-CeX_3 eutectics.

For Zr, we have already presented different types of $\text{Li}_{3-x}\text{M}_{1-x}^{3+}\text{M}_x^{4+}\text{X}_6$ halides that contain Zr^{4+} in our work and in **Table R4** and **Figure R5**.

For the effect of Br, we have presented that the structure transition from *hcp-T* to *ccp-M* can be realized by decreasing the ionic potential of X^- with the substitution of a larger anion. Two representatives are *hcp-T* typed Li_3YCl_6 and *ccp-M* typed Li_3YBr_6 . When Li^+ and Y^{3+} cations stack into the anion framework, the larger size of the Br^- anion will dilute the electronic cloud extension effect of cations, thus ensuring the high symmetry of the Br^- anion framework in the *ccp* arrangement. As displayed in the manuscript, the target sample of $\text{Li}_3\text{YCl}_{5.6}\text{Br}_{0.4}$ still remains the *hcp-T* structure, while further Br- substitution induces structural transitions to the *hcp-O* and finally the *ccp-M* phases at compositions of $\text{Li}_3\text{YCl}_{4.8}\text{Br}_{1.2}$ and $\text{Li}_3\text{YCl}_{4.5}\text{Br}_{1.5}$, respectively. The reported $\text{Li}_3\text{YCl}_3\text{Br}_3$ also possesses *ccp-M* structure^{10, 11}, which consistent well with our results. The similar structure and ionic conductivity evolution can be further confirmed in the $\text{Li}_3\text{ErCl}_{6-x}\text{Br}_x$, $\text{Li}_3\text{YCl}_3\text{Br}_3$ ²¹, and $\text{Li}_3\text{YbCl}_{6-x}\text{Br}_x$ halides as presented in **Figure R7**. All these samples consistent with the structural design rules and the structure transition from *hcp-T* (or *hcp-O*) to *ccp-M* can be realized by decreasing the ionic potential of X^- with the substitution of a larger anion Br^- instead of Cl^- in the $\text{Li}_3\text{MCl}_{6-x}\text{Br}_x$ systems.

Figure R7. (a) Analysis of the cationic polarization factor of $\text{Li}_3\text{ErCl}_{6-x}\text{Br}_x$ and $\text{Li}_3\text{YbCl}_{6-x}\text{Br}_x$ halides. (b) XRD pattern of $\text{Li}_3\text{HoCl}_3\text{Br}_3$, $\text{Li}_3\text{YCl}_3\text{Br}_3$, $\text{Li}_3\text{ErCl}_{1-x}\text{Br}_x$, and $\text{Li}_3\text{YbCl}_{6-x}\text{Br}_x$ halides. (c) The ionic conductivity evolution of the $\text{Li}_3\text{ErCl}_{1-x}\text{Br}_x$ and $\text{Li}_3\text{YbCl}_{6-x}\text{Br}_x$ halides at 25 °C.

In addition, **Figure R8** also presents that most the reported Br-contained halides follows our proposed structural design rules.

Figure R8. Cationic polarization factor of the Br-contained halides.

We further added these tables and figures in the revised manuscript and supporting information and highlighted.

Referee: #2

Comments to the Author

The present work deals with the systematic investigation of novel compositions of halide-based Li-ion conductors. Using a cationic polarization factor, the authors reveal compositions with increased ionic conductivity. The work presents a novel approach for the design and improvement of halide ion conductors and can be accepted for publication after minor revision. A point-by-point list of remarks is given in the following.

Response: Many thanks for your strong recommendation for publishing!

1. At the beginning of page 2, there is an error in the citing literature. [...]different Li⁺ migration properties. Please correct this.

Response: Many thanks for the suggestions. We have corrected this error in the revised manuscript and marked it yellow.

2. In Figure 1d the crystallographic axis should be included to get a better understanding of the structural changes.

Response: Many thanks for the suggestions. We have added the crystallographic axis in Figure 1d in the revised manuscript and marked it yellow.

3. As the cationic polarization factor is relatively new in the field of solid electrolytes, can the authors extend the explanation of how it was calculated (see page 4).

Response: Many thanks for the comments. The cationic polarization factor here is motivated by the Goldschmidt's tolerance factor that proposed for perovskite materials, which is calculated from the ratio of the ionic radii.¹²⁻¹⁴ In our case, the basic descriptor is also the radius ratio rule. For Li_aMX_b type halides based on octahedral sublattices, according to the law of ionic packing, they can be differentiated from other halides based on the radius ratio of cation to anion ($k = r^+/r^-$). Two critical values of 0.414 (octahedral factor) and 0.732 (cubic factor) are obtained from the geometric information as presented in **Fig. 1 c,d**. The octahedral sublattice structure can be theoretically stable when $0.414 < k < 0.732$. Thus, the two critical lines calculated from the radius ratio of

multivalent cations and halide ions distinguish the Li_aMX_b type halides from others (**Fig. 1e**). However, as we mentioned in the manuscript, this descriptor only accounts for the difference in cation/anion radii, whereas can not predict the structure of halides with the same metal cations and anions (*e.g.* *hcp-T* typed Li_3HoCl_6 and *hcp-O* typed $\text{Li}_{2.727}\text{Ho}_{1.091}\text{Cl}_6$). Therefore, the ionic potential of ions was further considered for the design rules. Here, the structures of the discussed Li_aMX_b type halides are based on the anion sublattice, which originating from the basic LiX structure that can be seen as the distribution of small Li^+ into the X^- anion sublattice. When forming Li_aMX_b halides, M^{3+} cations are further added to the system. Thus, the total effect of stacking cations (Li^+ and M^{3+}) into X^- is considered based on both radii and their electric charges as below:

$$\tau = \frac{\Sigma\Phi_{\text{cations}}}{\Sigma\Phi_{\text{X}}} = \frac{\Phi_{\text{Li}} + \Sigma\Phi_{\text{M}}}{\Sigma\phi_{\text{X}}}$$

4. What does “weighted ionic potential of lithium ions” mean? Please explain this in more detail.

Response: Many thanks for the suggestions. The weighted ionic potential of lithium ions Φ_{Li} represents the molar content proportionally ionic potential of lithium ions, which is defined as $\Phi_{\text{Li}} = n_{\text{Li}} \times I_{\text{Li}}$. Taking Li_3YCl_6 as an example, the ionic potential of Li^+ (I_{Li}) is 1.1111 as shown in Supplementary Table 3, the molar content of lithium ions in Li_3YCl_6 is 3, thus Φ_{Li} is 3.3333 by multiplying 1.1111 by 3. To make it clearer, we have revised the “weighted ionic potential of lithium ions” to “molar content proportionally ionic potential of lithium ions” in the revised manuscript and marked it yellow.

5. Please add a paragraph into the conclusion section discussing the possibility to transfer such concept of the cationic polarization factor to other classes of ion conductors.

Response: Many thanks for the suggestions. As suggested, we further add a paragraph into the conclusion section discussing the possibility of application of the cationic polarization factor to other classes of ion conductors as below:

From the viewpoint of crystal structures, the halide SSEs discussed in our work are mainly based on octahedral coordination similar to the studies of various tolerance factors proposed for perovskite materials^{13, 14}. Moreover, the cationic polarization factor proposed here is based on the

ion stacking in ionic crystals. For other classes of ion conductors (such as sulfides, polyanionic oxides), most of these materials contain structural frameworks that involve covalent bonds, such as PS_4^{3-} , PO_4^{3-} , and SiO_4^{4-} . The crystal structures of these ion conductors possess various coordination conditions (four-coordination for PS_4^{3-} , eight-coordinated LaO_8^{13-} , etc.)^{15, 16}, which are totally different from that of halide SSE discussed here. Thus, the cationic polarization factor is not suitable for these materials. Precisely because of the structural and chemical distinctions, the Li^+ migration through these frameworks will lead to the different phenomenon and diffusion mechanisms in halide SSEs. For ion conductors with specific crystal structures, it's still possible to investigate the formability by similar rules. We also hope that this work will be helpful to the modification of tolerance factor for perovskite materials by further considering ionic charges rather than just ionic radii effect.

Referee: #3

Comments to the Author

Halides are currently a hot topic for SEs. In this manuscript, the authors try to give and prove the structural regulation and evolution of halide SEs by using the cationic polarization factor, etc. They provide a predicted model of Li-M-X phase diagram, which is highly related to the Li-ion transport capability. This work is important for the halide design and may be a good reference for other systems. It is also interesting that they provide over 10 Li-halide electrolytes possessing high conductivities over 1 mS/cm. I think the prediction is reasonable. Therefore, I recommend to publish the manuscript in Nature Communication.

Response: Many thanks for your strong recommendation for publishing!

1. Why do you use Shannon crystal radii rather than the Pauling radius of ions, as we know Pauling radius is also widely used? What are the differences for them and the scope of their respective application? I suggest to give more detailed information in the manuscript.

Response: Many thanks for the suggestions. Ionic radius is not a constant value for a particular ion but it changes with oxidation state, coordination environment, chemical composition, and orbital

configurations. Pauling used effective nuclear charge to proportion the distance between ions into anionic and a cationic radius. For example, The Pauling ionic radius of O^{2-} ion is 140 pm. Shannon gives different radii for different coordination numbers, oxidation states, and for high and low spin states of the ions. He also includes data are referred to "crystal" ionic radii, which are believed that crystal radii correspond more closely to the physical size of ions in a solid. In our work, the discussed Li_aMX_b halides are highly dependent on the coordination information and crystal structures. Thus, Shannon crystal ionic radii are used in our case rather than Pauling radius of ions.

Based on the reviewer's comments, we have further declared that in the revised manuscript and marked it yellow as below:

"The radii used here were obtained based on Shannon crystal ionic radii which are closer to the physical size of ions in a solid."

2. The structure model is based on the factors, e.g., ion polarization. I wonder how much influence the electronic structure of different elements has on this model?

Response: Many thanks for the suggestions. The structural model in our work is mainly related to the Li_aMX_b halides with ionic bonds. The ion polarization discussed here has already consider the electronic structure of different elements since the ion polarization are highly related to ion charges of different elements. In this context, the factor of electronic structure has already been considered in this model. Moreover, as the Li_aMX_b halides are based on ionic bonds, the model here likes stacking Li^+ , M^{3+} , M^{4+} balls into the X^- anion sublattice. Therefore, the directional properties of covalent bond due to the different electronic structure of elements are not considered in this model.

3. The phase structures of hcp-T, hcp-O, and ccp-M show different ion conduction rates. Why and what are the rules about them? Does it mean that the symmetry of the halide structure and/or the arrangement of the anion framework have a significant influence on ion transport?

Response: Many thanks for the suggestions. For Li_aMX_b SSEs with a *ccp-M* structure, the pathways of Li^+ conduction are connected via tetrahedral interstitial sites between edge-sharing octahedral sites in all three directions, forming a three-direction isotropic diffusion network. For

the hcp anion arrangement (*hcp-O* and *hcp-T*), Li^+ transports along the *ab*-plane are via tetrahedral interstitial sites, which is similar to the *ccp-M* structure. Along the *c*-axis, the diffusion paths are directly connected between neighboring octahedral sites. Due to the different Li^+ pathways, the ion conduction rates within these three structures are different and usually higher ionic conductivities can be achieved for the *ccp-M* typed and *hcp-O* typed Li_aMX_b halides. Thus, both the symmetry of the halide structure and the arrangement of the anion framework have a significant influence on ion migration behavior.

4. The oxychloride-based and UCl_3 -type SEs e.g., Li-La-Ta-Cl have been reported recently, how about to use this phase diagram model in these novel electrolytes? Have you considered to use the model in other systems say, sulfide SEs?

Response: Many thanks for the suggestions.

From the viewpoint of crystal structures, the halide SSEs discussed in our work is mainly based on octahedral coordination similar to the studies of various tolerance factors that proposed for perovskite materials^{13, 14}. Recently reported oxychloride-based and UCl_3 -type materials possess different crystal structure. For instance, some of the oxychloride SSEs are almost amorphous¹⁷ while still show high ionic conductivities. The UCl_3 -type SSEs are mainly based on nine-coordinated cation sublattice^{18, 19}, which is totally different from our case. For other classes of ion conductors (such as sulfides, polyanionic oxides), most of these materials contain structural framework that involving covalent bonds, such as PS_4^{3-} , PO_4^{3-} , SiO_4^{4-} . The crystal structures of these ion conductors possess various coordination conditions (four-coordination for PS_4^{3-} , eight-coordinated LaO_8^{13-} , etc.)^{15, 16}, which are totally different from that of halide SSE discussed here. Thus, the cationic polarization factor is not suitable for these materials.

5. The references for the definition of ionic potential should be cited. Regarding the formula $\Phi\text{Li} = n\text{Li} \times \phi\text{Li}$, ϕLi is not defined.

Response: Many thanks for the suggestions. We have added the references for the definition of ionic potential in the revised manuscript and marked it yellow. For the formula, we have corrected

it to as $\Phi_{Li} = n_{Li} \times I_{Li}$ to make it consistent with the definition of ionic potential (I) in the manuscript, here I_{Li} represents ionic potential of lithium ion. We also correct this in the revised manuscript.

Reference:

1. J. Fu, S. Yang, J. Hou, L. Azhari, Z. Yao, X. Ma, Y. Liu, P. Vanaphuti, Z. Meng and Z. Yang, *Journal of Power Sources*, 2023, **556**, 232465.
2. E. van der Maas, T. Famprakis, S. Pieters, J. P. Dijkstra, Z. Li, S. R. Parnell, R. I. Smith, E. R. van Eck, S. Ganapathy and M. Wagemaker, *Journal of Materials Chemistry A*, 2023, **11**, 4559-4571.
3. H. Kwak, D. Han, J. P. Son, J. S. Kim, J. Park, K.-W. Nam, H. Kim and Y. S. Jung, *Chemical Engineering Journal*, 2022, **437**, 135413.
4. H. Wang, Y. Li, Y. Tang, D. Ye, T. He, H. Zhao and J. Zhang, *ACS Applied Materials & Interfaces*, 2023, **15**, 5504-5511.
5. G. Xu, L. Luo, J. Liang, S. Zhao, R. Yang, C. Wang, T. Yu, L. Wang, W. Xiao and J. Wang, *Nano Energy*, 2022, **92**, 106674.
6. J. Park, D. Han, H. Kwak, Y. Han, Y. J. Choi, K.-W. Nam and Y. S. Jung, *Chemical Engineering Journal*, 2021, **425**, 130630.
7. K.-H. Park, K. Kaup, A. Assoud, Q. Zhang, X. Wu and L. F. Nazar, *ACS Energy Letters*, 2020, **5**, 533-539.
8. Q. Shao, C. Yan, M. Gao, W. Du, J. Chen, Y. Yang, J. Gan, Z. Wu, W. Sun and Y. Jiang, *ACS Applied Materials & Interfaces*, 2022, **14**, 8095-8105.
9. L. Zhou, T. Zuo, C. Li, Q. Zhang, J. r. Janek and L. F. Nazar, *ACS Energy Letters*, 2023, **8**, 3102-3111.
10. E. van der Maas, W. Zhao, Z. Cheng, T. Famprakis, M. Thijs, S. R. Parnell, S. Ganapathy and M. Wagemaker, *The Journal of Physical Chemistry C*, 2023, **127**, 125-132.
11. Z. Liu, S. Ma, J. Liu, S. Xiong, Y. Ma and H. Chen, *ACS Energy Letters*, 2020, **6**, 298-304.
12. C. Shi, C. H. Yu and W. Zhang, *Angewandte Chemie*, 2016, **128**, 5892-5896.
13. C. J. Bartel, C. Sutton, B. R. Goldsmith, R. Ouyang, C. B. Musgrave, L. M. Ghiringhelli and M. Scheffler, *Science advances*, 2019, **5**, eaav0693.
14. M. R. Filip and F. Giustino, *Proceedings of the National Academy of Sciences*, 2018, **115**, 5397-5402.
15. C. Wang, K. Fu, S. P. Kammampata, D. W. McOwen, A. J. Samson, L. Zhang, G. T. Hitz, A. M. Nolan, E. D. Wachsman and Y. Mo, *Chemical reviews*, 2020, **120**, 4257-4300.
16. K. Homma, M. Yonemura, T. Kobayashi, M. Nagao, M. Hirayama and R. Kanno, *Solid State Ionics*, 2011, **182**, 53-58.
17. S. Zhang, F. Zhao, J. Chen, J. Fu, J. Luo, S. H. Alahakoon, L.-Y. Chang, R. Feng, M. Shakouri and J. Liang, *Nature Communications*, 2023, **14**, 3780.
18. Y.-C. Yin, J.-T. Yang, J.-D. Luo, G.-X. Lu, Z. Huang, J.-P. Wang, P. Li, F. Li, Y.-C. Wu and T. Tian, *Nature*, 2023, **616**, 77-83.
19. J. Fu, S. Wang, J. Liang, S. H. Alahakoon, D. Wu, J. Luo, H. Duan, S. Zhang, F. Zhao and W. Li, *Journal of the American Chemical Society*, 2022, **145**, 2183-2194.

REVIEWERS' COMMENTS

Reviewer #1 (Remarks to the Author):

The authors have addressed all comments and suggestions. I recommend acceptance of this manuscript.